# Histopathological and Behavioral Impairments in Zebrafish (*Danio rerio*) Chronically Exposed to a Cocktail of Fipronil and Pyriproxyfen

**DOI:** 10.3390/life13091874

**Published:** 2023-09-06

**Authors:** Madalina Andreea Robea, Adriana Petrovici, Dorel Ureche, Mircea Nicoara, Alin Stelian Ciobica

**Affiliations:** 1Doctoral School of Biology, Faculty of Biology, “Alexandru Ioan Cuza” University of Iasi, 700505 Iasi, Romania; madalina.robea11@gmail.com; 2Department of Preclinics, University of Life Sciences, 700490 Iasi, Romania; 3Regional Center of Advanced Research for Emerging Diseases, Zoonoses and Food Safety, 700490 Iasi, Romania; 4Department of Molecular Biology, Histology and Embryology, Faculty of Veterinary Medicine, University of Agricultural Science and Veterinary Medicine “Ion Ionescu de la Brad”, 700489 Iasi, Romania; 5Faculty of Sciences, Department of Biology, Ecology and Environmental Protection, University “Vasile Alecsandri”, 600115 Bacau, Romania; 6Department of Biology, Faculty of Biology, “Alexandru Ioan Cuza” University of Iasi, 700505 Iasi, Romania; 7Doctoral School of Geosciences, Faculty of Geography and Geology, “Alexandru Ioan Cuza” University of Iasi, 700505 Iasi, Romania; 8Academy of Romanian Scientists, 050094 Bucharest, Romania; 9Center of Biomedical Research, Romanian Academy, 700506 Iasi, Romania

**Keywords:** pesticide, fipronil, pyriproxyfen, behavior, histopathological changes, zebrafish

## Abstract

Background: Lately, the high incidence of pesticide usage has attracted everyone’s interest due to the serious effects produced. Fipronil (FIP) is a phenylpyrazole compound that acts on the insect’s GABA neurotransmitter by inhibiting its activity. Moreover, the literature reports highlight its implication in neurodevelopmental abnormalities and oxidative stress production in different organisms. Similarly, pyriproxyfen (PYR) is known to affect insect activity by mimicking the natural hormones involved in the maturation of the young insects. The aim of the present study was to investigate the impact of the mixture of these pesticides on the tissues and behavior of zebrafish. Methods: To assess the influence of this cocktail on zebrafish, three groups of animals were randomly selected and exposed to 0, 0.05, and 0.1 mg L^−1^ FIP and PYR mixture for five days. The fish were evaluated daily by the T-maze tests for locomotor activity and the light–dark test and recordings lasted four min. The data were quantified using the EthoVision software. Results: Our results indicated significant changes in locomotor activity parameters that showed increased levels following exposure to the mixture of FIP and PYR. On the other hand, the mixture also triggered anxiety in the zebrafish, which spent more time in the light area than in the dark area. In addition, mixture-induced histological changes were observed in the form of numerous hemosiderin deposits found in various zebrafish tissues. Conclusions: The current findings indicate that the mixture of FIP and PYR can have considerable consequences on adult zebrafish and may promote or cause functional neurological changes in addition to histological ones.

## 1. Introduction

Chemicals such as pesticides are frequently used in agricultural production to prevent or manage pests, weeds, and other plant pathogens in order to reduce or eliminate yield losses while maintaining good product quality [1,2,3]. Despite the fact that pesticides are designed to have a low impact on human health and the environment, major concerns have been raised about the health risks associated with occupational exposure and residues found in food and drinking water [2,4].

Environmental exposure affects the human population mainly through eating contaminated food and drinking contaminated water as well as applying pesticides in the home or living near treated fields. Consequently, risk assessment and prevention of exposure to pesticides is a complex process, especially when several limitations are considered, such as differences in time and level of exposure, chemical structure and specific toxicity of the pesticide, mixtures or cocktails applied, and, last but not least, the geographical and climatic characteristics of the areas where they are used [2,4].

Fipronil (FIP) is a broad-spectrum insecticide from the phenylpyrazole family (Figure 1A) [5,6,7]. In the past few years, FIP has been frequently highlighted in research publications as one of the adopted pest control chemicals whose effects have not been sufficiently investigated. According to the National Pesticide Information Center (NPIC), FIP was first registered in 1996 as a broad-spectrum insecticide against ants, beetles, cockroaches, fleas, ticks, termites, thrips, rootworms, weevils, and other insects. 

Due to its massive use and negative impact on bee populations, FIP has been restricted in several countries. Consequently, in 2013, FIP was banned in Europe for agricultural applications; nowadays, it is only used in several veterinary products [8]. On the other hand, unlike Europe, Asia, and South America, the United States of America continues to use FIP for both agriculture and domestic purposes [9]. For instance, discharge from 12 facilities after a plant treatment reported fiproles (FIP and its degradation products) concentrations ranging from 0.3 to 112.9 ng L^−1^, exceeded the United States Environmental Protection Agency (USEPA) aquatic invertebrate life threshold established for chronic exposure to FIP (11 ng L^−1^) in over 67% of cases [10]. 

The mechanism of action is not fully characterized; everything that is known so far is due to the studies carried out on insects which showed a significant toxicity compared to other organisms. Its main pathway involves binding to GABA_A_-gated chloride channels and further blocking the inhibitory action of GABA_A_ in the central nervous system (CNS) [11,12,13]. FIP has been linked to a variety of negative consequences in humans and other species, such as cytotoxicity, neurotoxicity, hepatotoxicity, and reproductive issues. In rodent studies, FIP has been linked to important alterations in growth, development, and genotoxic and cytotoxic disturbances in addition to those in the reproductive domain [11,14,15,16]. Dermal exposure of Wistar rats to 70, 140, and 280 mg kg^−1^ FIP after 3 hours (h) elicited increases in rearing, freezing, and grooming behavior assessed by the open field test [17]. Chronic treatment of male Wistar rats (6 weeks) with 4.85 mg kg^−1^ FIP showed impairments in spatial learning and memory performance assessed by Barnes maze. In addition, decreases in antioxidant system activity and dopamine and elevated levels in malondialdehyde (MDA) and serum corticosterone were obtained [18]. Several of these observed effects, such as reduced food intake and weight gain, were also observed in bobwhite quail (*Colinus virginianus*) after a single dose of 37.5 mg kg^−1^ FIP and measured 8, 24, 48, 72, and 96 h after administration [19].

A recent study on the impact of FIP on freshwater mussels (*Unio delicatus*) showed that exposure to 0.264 and 0.528 mg L^−1^ for 48 h and 7 days can lead to significant alterations in gills and digestive tissues, such as degeneration or even necrosis [20]. Furthermore, the negative impact of FIP exposure was also obtained for liver cells that had major ultrastructural changes characterized by cytoplasmic disorganization, fat accumulation, and increased numbers of mitochondria, rough endoplasmic reticulum and Kupffer cells after 15, 25, and 50 mg kg^−1^ administered intraperitoneally to female Swiss mice (*Mus musculus*) [21]. 

In addition, evidence of exposure to FIP in non-target organisms has been reported, particularly when it was used for crop protection. Related to this claim, a 2012 study evaluating the effect of FIP (0.65 mg L^−1^) in rice-field circumstances on common carp (*Cyprinus carpio*) determined that 90 days of exposure to FIP can cause alterations in fish metabolism in terms of toxicological profile. Changes in superoxide dismutase (SOD), catalase (CAT) activity, and levels of lipid peroxidation and protein carbonyl production were recorded but there was no impact on growth rate or survival [22]. 

Another chemical introduced to control insects is pyriproxyfen (PYR) (Figure 1B). The first time it was brought up was in 1991 when experts were trying to fight sweet potato whiteflies (*Bemisia tabaci*) and greenhouse whiteflies (*Trialeurodes vaporariorum*) in cotton and tomato crops. Exposure to different doses of PYR had significant effects on reproductive status: 48 h exposure of female whiteflies to 5 mg L^−1^ PYR resulted in suppression of egg viability [23]. A few years later, in 1996, the US introduced PYR for the same reason: to protect cotton crops. PYR is known as an endocrine disruptor that mimics the juvenile hormone needed by insects to molt from the immature stage to the adult stage [24]. Moreover, it has been shown that exposure to PYR in insects can affect the reproductive process [24]. Due to its ability to persist in the environment (low solubility and hydrophobicity), precautions have often been required [24,25].

Widely used as an insect growth regulator and because it is considered a non-toxic compound, in 2006 the World Health Organization recommended the use of 0.01 mg L^−1^ PYR for the disinfection of drinking water, especially for the control of dengue fever [26]. The same recommendation was followed in 2014 by the Brazilian Ministry of Health to fight mosquitoes [27]. Although there are studies that mention the low toxicity of PYR to other species, recent findings argue otherwise. For instance, oral administration of 20, 40, 100, 200, 320, 600, and 1200 mg kg^−1^ PYR for 28 days led to a decrease in the body weight of mice [28]. This is similar to the study of Mehrnoush et al. who tested 1000, 2000, and 4000 mg kg^−1^ PYR in rats [29]. A reduction in feeding capacity was also recorded for juvenile platy (*Xiphophorus maculatus*) after intake of 2.5 and 5 µg L^−1^ PYR [30]. Several findings were made related to the influence of PYR on zebrafish where the main finding identified was related to neurological changes, namely impairment of acetylcholinesterase activity, oxidative stress products, calcium transport malfunction, and mitochondrial damage. Several findings were made related to the influence of PYR on zebrafish, where the main finding identified was related to neurological changes, namely impairment of acetylcholinesterase activity, oxidative stress products, calcium transport malfunction, and mitochondrial damage [31].

A large percentage of pesticides in use today are potentially neurotoxic, affecting the nervous systems of target organisms, among other organs and component systems. So, the effect of these compounds can also be observed at the histological level [32,33,34]. For example, FIP has been reported to induce multiple tissue changes in various animal models, with a more pronounced effect on the central nervous system (CNS). In rodent models, FIP has been linked to altered kidney, liver, and brain anatomy and function following insecticide administration [35,36,37,38]. After exposure of common carp (*Cyprinus carpio*) to a dose range between 0.02 and 0.10 mg L^−1^ FIP for 12 days, histopathological changes were identified in the gills, liver, kidneys, and intestine [39]. Exposure to 1.66 µg mL^−1^ PYR for 96 h caused significant alterations in zebrafish heart anatomy, including heart muscle thinning, hyperemia, and pericardial edema [40]. The influence of PYR at the histological level was also demonstrated in female mice after the administration of 30, 100, 300, and 1000 mg kg^−1^ which led to an increase in the number of glial cells, vacuolization, and degenerated neurons [41]. 

For the present study, one of the most well-known model organisms was chosen, namely zebrafish or *Danio rerio*, which is frequently used in research [42,43,44]. Its attractiveness for this field is supported by several advantages, including a high genome similarity to humans (about 70%), transparent larval stage, high number of eggs, and low cost of care [44]. Along with the advantages mentioned above, the successful adaptation of behavioral tests and the ease of genetic manipulation on them has led to their success compared to rodents [45,46,47].

Therefore, this experimental protocol aimed to test two doses of FIP and PYR mixture in chronic exposure in zebrafish. The first goal was to evaluate the impact of the mixture on zebrafish locomotor activity parameters and anxious behavior. The second one was to investigate the effect of this mixture at the histological level. From what is known so far, the impact of this mixture has not been studied in other works, except for one study belonging to this research group [48].

## 2. Materials and Methods

### 2.1. Fish Maintenance

The test organisms, adult wild-type zebrafish (*Danio rerio*), were purchased from a local supplier in a total number of 150 with a sex ratio of 1:1 (female:male). The fish were accommodated in the facility for a period of 3 weeks before the start of the experimental study. In addition, it was ensured that all conditions for maintaining a healthy environment were met in accordance with the recommendations of the European Commission regulations for housing and care of animals used for experimental and other scientific purposes, in addition to those of Avdesh et al. and Aleström. et al. [49,50,51]. To ensure environment-specific conditions, daily water quality measurements of temperature, pH, salinity, conductivity, and ammonia were performed (Table 1). The medium was changed daily in the housing and experimental tanks to avoid the accumulation of metabolic wastes such as ammonia. In addition to water quality, the illumination period was also set to a circadian rhythm of 14:10 h (light: dark) by an automatic timer that was started at 8 a.m. and off at 10:00 p.m. The fish were fed twice a day with flakes from TetraMin products (Herrenteich 78 49324 Melle, Germany).

### 2.2. Chemicals

The mixture of FIP and PYR used in this study was purchased in liquid form from a veterinary store. In addition to these two compounds (67.5 mg FIP + 67.5 mg PYR), the product also contained 0.3 mg butylhydroxyianisol, 60 mg benzyl alcohol, and 0.15 mg butylhydroxytoluene. For this study, the doses chosen for testing were 0.05 and 0.1 mg L^−1^ (based on previous unpublished data). These were obtained by dissolving a certain amount of the prepared stock solution in the zebrafish medium.

### 2.3. Experimental Design

Fish were randomly transferred from the housing aquarium to the experimental tanks and allowed to habituate to the new space. Three experimental tanks were established: control (*n* = 10), 0.05 mg L^−1^ FIP and PYR (*n* = 10), and 0.1 mg L^−1^ FIP and PYR (*n* = 10) for a 5 day exposure. At the end of accommodation period, the initial behavior of each fish was tested in the T-maze. Specific parameters for locomotor activity and anxiety were assessed after a 4-min session (each test is described in the behavioral tests section). For the treatment phase, the pesticide mixture was freshly prepared before exposure and changed every 24 h until the end of the study. To investigate the impact of FIP + PYR on zebrafish behavior, locomotor activity test and light–dark test were performed daily. Finally, fish were killed by immersion in ice-cold water for at least 5 min after opercular movement stopped and then stored for histological analysis. The study had two more replicates. 

### 2.4. Behavioral Tests

#### 2.4.1. Locomotor Activity Test

An adapted T-maze was used to measure specific locomotor activity parameters of zebrafish before and after treatment. It was divided into 3 arms (left, right, and center) and had a starting point at the end of the central arm (Figure 2). For the analysis of fish locomotion, a series of parameters were quantified: total distance traveled, average velocity, active time, and time spent in inactivity. In addition to these, specific parameters were measured to describe the swimming pattern: angular velocity, meander, and turning angle. The total distance traveled represents the total distance swam by fish in the T-maze (cm) during the trial. Average velocity refers at how fast the fish is moving (cm s^−1^). Counter-clockwise rotations indicate circular movement of the fish to the left. The angle of turn defines the angle at which the fish’s head turns during a turn (°). Tests were performed every day and all parameters were quantified for a period of 4 min per sample. Heatmaps are also added to this data.

#### 2.4.2. The Light–Dark Test

This test aimed to assess the fish’s anxiety level when it encounters two different environments: light and dark. To perform this test, the T-maze apparatus was adapted by placing a wall and transforming it into a rectangular shape (10 cm height × 50 cm length × 10 cm width) and designed according to the protocol of Araujo et al. [52]. It was further divided by two walls into three areas: light, dark, and decision point (Figure 3). Before the session started, each fish was placed at the decision point for 30 s to accommodate (Figure 3a). After this time, the walls for the light and dark areas were removed and the fish could swim freely between these two areas (Figure 3b). The session lasted 4 min, and the activity was recorded, like the locomotor activity test, through the camera located above the maze. To validate anxiety behavior, several parameters were recorded such as the distance traveled, average speed, and time spent in each zone of the maze.

### 2.5. Assessment of Histological Changes

From the control group and from each tested group, 10 fish were euthanized by immersion in cold water, fixed in 4% PFA, decalcified in EDTA 0.35 M—pH 8, embedded in paraffin, and cut in 12 µm tissue sections. The entire central nervous system was cut and stained with Luxol Fast–Blue–Cresyl Violet (LFB-Cr. V) for the examination of the gray and white matter. Image acquisition was made with a Motic Easy Scan Pro 6 scanner system (Kowloon City, Kowloon, Hong Kong, Motic). 

### 2.6. Statistical Analysis

Behavioral data were first assessed for normal distribution and confirmed by the Shapiro–Wilk test. To observe the differences between the concentrations of FIP + PYR and the control group, one-way ANOVA was followed by Tukey’s post hoc test. Differences were considered significant at *p* < 0.05. Data are expressed as mean ± standard error of the mean (S.E.M.). Microsoft Package Excel files (Microsoft Office Professional Plus 2021) and OriginPro software (OriginLab Corporation 2021) were used for data selection, statistical analysis, and graphical presentation.

### 2.7. Ethical Approval

Animals were strictly maintained and treated in accordance with the EU Commission Recommendation (2007) on guidelines for the accommodation and care of animals used for experimental and other scientific purposes, Directive 2010/63/EU of the European Parliament and of the Council of 22 September 2010, on the protection of animals used for scientific purposes [49,53]. This experiment was approved by the Ethics Committee of the Faculty of Veterinary Medicine, University of Agricultural Sciences and Veterinary Medicine Iasi, with registration number 749/04.07.2019.

## 3. Results

### 3.1. Impaired Locomotor Activity after Chronic Exposure to Pesticide Mixture 

There were no significant differences recorded for the control group (*p* = 0.06 ANOVA, Tukey) and the group exposed to the 0.05 mg L^−1^ mixture (*p* = 0.35 ANOVA, Tukey) regarding the total distance traveled during the experimental sessions (Table 2). Thus, the data for this parameter recorded the highest peak on the fourth day of the study for the 0.1 mg L^−1^ mixture group compared to the initial behavior: 1437.5 ± 112.9 vs 895.2 ± 74.3 cm (*p* = 0.0016 ANOVA, Tukey). At the end of treatment, its activity was still increased compared to the PTR phase: 1318.3 ± 87.1 vs 895.2 ± 74.3 cm (*p* = 0.023 ANOVA, Tukey) (Figure 4).

Similar to the above parameter, the average velocity did not show important changes among the individuals from the control and 0.05 mg L^−1^ FIP + PYR groups (*p* = 0.07 and *p* = 0.35 ANOVA, Tukey) compared to the 0.1 mg L^−1^ FIP + PYR group (Table 3). It was also indicated that there were elevated averages for D_4 (5.9 ± 0.4 cm s^−1^, *p* = 0.0017 ANOVA, Tukey) and D_5 (5.4 ± 0.3 cm s^−1^, *p* = 0.024 Tukey) in comparison to the initial behavior (3.7 ± 0.3 cm s^−1^) (Figure 5).

In addition, counterclockwise rotations and turning angles were measured during the locomotor activity test. For both parameters, the control group did not show high variability between the baseline behavior and the sham treatment period (*p* = 0.20 and *p* = 0.41 ANOVA, Tukey). Regarding the experimental groups exposed to the pesticide mixture (0.05 and 0.1 mg L^−1^ FIP + PYR), a tendency to increase the number of counter-clockwise rotations was observed but without significant disturbances (*p* = 0.47 ANOVA, Tukey) (Figure 6a). Another indicator of possible changes in locomotor activity is the turning rate of the fish. Also, the results showed that the pesticide mixture did not trigger any significant effect on the turning angle for both tested doses (*p* = 0.79 and *p* = 0.77 ANOVA, Tukey) (Figure 6b). More data are available in Table 4. 

### 3.2. Increased Anxiety Level of Zebrafish

The total distance recorded during the light–dark test showed significant differences between all pesticide groups (Table 5). No significant changes were observed for the control group (*p* = 0.08 ANOVA, Tukey). Exposure to 0.05 mg L^−1^ FIP + PYR increased the mean distance traveled by fish, with significant data for D_1 (2874.6 ± 231.1 cm, *p* = 0.01 ANOVA, Tukey), D_4 (3273.1 ± 252.3 cm, *p* = 0.003 ANOVA, Tukey), and D_5 (3046.8 ± 119.6 cm, *p* = 0.02 ANOVA, Tukey) compared to baseline behavior (1982.6 ± 180.7 cm). Compared to the 0.05 mg L^−1^ FIP + PYR group, which showed increased activity levels during chronic treatment, the 0.1 mg L^−1^ FIP + PYR group had a similar trend, but not as many changes in distance values except D_2 (3723.5 ± 345.5 cm vs. PTR: 2515.7 ± 175.6 cm, *p* = 0.01 ANOVA, Tukey) (Figure 7).

As shown in Figure 8 and in Table 6, both doses of pesticides also affected the average speed of the fish. The control group had ups and downs with no significant changes in their activity. Conversely, the 0.05 mg L^−1^ FIP + PYR group showed increased mean velocities in D_1 (12.3 ± 0.9 cm s^−1^, *p* = 0.01 ANOVA, Tukey), D_4 (13.9 ± 1.1 cm s^−1^, *p* = 0.005 ANOVA, Tukey), and D_5 (12.9 ± 0.4 cm s^−1^, *p* = 0.03 ANOVA, Tukey) when compared to the PTR phase (8.71 ± 0.7 cm s^−1^). Regarding the other group exposed to 0.1 mg L^−1^ FIP + PYR, the mean velocity showed a high value for D_2 (15.9 ± 1.4 cm s^−1^, *p* = 0.01 ANOVA, Tukey) compared to the PTR period (10.7 ± 0.7 cm s^−1^) (Figure 8).

The time spent by the groups in a certain area of the experimental apparatus was quantified for the light, dark, and decision point compartments, as represented in Figure 8. There were no significant changes in the activity of the control and 0.05 mg L^−1^ FIP + PYR groups for any zone of the maze (*p* = 0.11 and *p* = 0.49 ANOVA, Tukey for dark zone, *p* = 0.66 and *p* = 0.52 ANOVA, Tukey for light zone, *p* = 0.55 and *p* = 0.49 ANOVA, Tukey for the decision point zone) (Figure 9). Thus, fish exposed to 0.05 mg L^−1^ FIP + PYR showed increased time spent in the light zone for D_4 (147.9 ± 19.9 s, *p* = 0.02 ANOVA, Tukey) and D_5 (168.4 ± 24.6 s, *p* = 0.02 ANOVA, Tukey) compared to baseline behavior (98.8 ± 10.2 s) (Figure 9). In Table 7, all the experimental data means ± S.E.M. are presented.

### 3.3. Histological Alterations

The impact of the mixture of FIP and PYR on the zebrafish CNS was analyzed in the following brain regions: olfactory bulbs, telencephalon, diencephalon and mesencephalon, rhombencephalon, and medulla spinalis. In the most rostral region of the adult brain, cell density appeared altered with fewer perikaryon in FIP + PYR-exposed groups compared with the control group (Figure 10(1A–1C)). Vasodilatation was evident in both treated groups and infiltrations were visible. No major differences were observed between the 0.05 and 0.1 mg L^−1^ FIP + PYR groups (Figure 10).

In the telencephalon, numerous blood vessels were found, some of which were ectatic in both treatment groups, as well as blood cell infiltration (Figure 10(2A–2C). In some neurons in the 0.1 mg L^−1^ FIP + PYR group, the nucleus was pyknotic and the cytoplasm was more eosinophilic, revealing the apoptotic effect of the mixture (Figure 10(2C)). 

Dilated blood vessels and leucocyte infiltration persisted in treatment groups in the major regions from diencephalon and mesencephalon (Figure 10(3A–3C)). The neuropil from the 0.1 mg L^−1^ FIP + PYR group was inhomogenously stained with Luxol Fast-Blue. The nerve fibers in the torus longitudinalis (TL) of the most exposed group were weakly stained. Central chromatolysis in large neurons in the oculomotor nucleus (NIII) was also distinguished (Figure 10(3C)). Dilated blood vessels and leukocyte infiltration persisted across treatment groups in major regions of the diencephalon and midbrain (Figure 10).

Regarding the rhombencephalon, mild infiltration and neuronal damage were observed, especially in the most exposed group of fish. The cerebellum had no obvious changes (Figure 10(4A–4C)). Perikaryon swelling in some motoneurons in the group exposed to 0.1 mg L^−1^ FIP + PYR was observed in the medulla (Figure 10(5C)). Also, intense vascularization was present in this area.

## 4. Discussion

This study analyzed the effect of the mixture of FIP and PYR on the tissues and behavior of zebrafish after five days of exposure. Both treatment groups showed higher values of the total distance traveled parameter during all behavioral testing sessions compared to baseline data. This effect was more pronounced for the 0.1 mg L^−1^ FIP and PYR group that recorded significant activities for D_4 and D_5. A similar trend was observed for the velocity parameter. These observations could support the hypothesis that FIP can initiate and maintain the body’s hyperactivity if the exposure period is prolonged. The results are in line with the study by Koslowski et al. where mice treated with FIP at 10 mg kg^−1^ for 43 weeks showed that long-term administration can cause changes in locomotor activity. They obtained higher values for distance traveled and time spent mobile after 43 weeks compared to the first session (18 weeks) assessed by the open field test [54]. This may be the result of FIP accumulating in the body with a considerable effect on the neurotransmitter GABA, which triggers these behavioral disturbances observed in most animal studies. For example, Reichel et al. showed that depletion of GABA-ergic neurons in the hippocampus of C57BL/6 mice can sustain the hyperactivity state [55]. Such disturbances in locomotor performance were not obtained by Gusso et al., who tested 0.125, 0.675, and 1.75 mg L^−1^ PYR on adult zebrafish, for 96 h [56]. 

Reduced levels of the neurotransmitter GABA are linked to attention deficits and hyperactivity in children. Under normal conditions, the binding of GABA to its receptors causes the channels to open, allowing chloride ions to move into the cell. This action decreases the excitability of the cell. When FIP or its metabolites bind to GABA_A_ receptors, it prevents the uptake of chloride ions, leading to neuronal hyperexcitability [57,58]. Additionally, the ability of PYR to disrupt the CNS was recently highlighted in a 2021 study that found inhibitory effects of 0.33 and 92.5 μmol L^−1^ PYR on acetylcholinesterase activity in male zebrafish [31]. 

The other two parameters assessed for swimming performance were counterclockwise rotations and turning angles. Zebrafish treated with FIP and PYR (0.05 and 0.1 mg L^−1^) showed no significant changes. Knowing that counterclockwise rotations are an indicator of abnormal swimming and suggestive of anxious behavior, in our case, both treated groups had an increased number of rotations compared to baseline data. In terms of the turning angle, which may support the theory of irregular movement, the mixture groups tended to have lower amplitudes of angles defined by low-angle turns (20 to 40°).

Regarding anxiety behavior, the experimental groups exposed to the mixtures of 0.05 and 0.1 mg L^−1^ showed an increase in the time spent in the light zone. The effect was more pronounced at the higher dose of the mixture compared to baseline behavior. In general, zebrafish adults tend to spend more time in the dark area, which is similar to the normal environment, compared to the larval stage, when they prefer the light area. In accordance with these results, the ability of FIP to enhance this type of behavior was validated for zebrafish larvae and adults. Based on the light–dark preference test, it was assessed that early exposure to 200 mg L^−1^ FIP can promote anxious behavior in zebrafish larvae, where avoidance of the dark area by the organism was considered as an indicator of an anxious state [59]. In contrast to the present findings, the Gusso et al. study demonstrated that exposure to PYR for 96 h could not induce anxiety-like behavior in adult zebrafish [56]. Although PYR may contribute to the onset and development of several cognitive deficits, changes in anxiety are not related to it. Thus, it can be assumed that FIP is solely responsible for this. Furthermore, dermal exposure to FIP was found to promote changes in emotional, fear, and exploratory activity according to a 2011 study in which Terçariol and Godinho assessed the behavior of Wistar rats [17].

Histopathological examination revealed congestion, inflammatory changes, and apoptosis in major brain regions in response to FIP + PYR mixtures. Fish in both treatment groups reacted to the substances in a similar way, with slightly more intense congestion and inflammation in the higher dose group. Neuronal impairment was present in all major CNS regions, such as vacuolization, central chromatolysis, and pyknotic nuclei. All of these are well-known signs of impending apoptosis. This may be a consequence of the presence of FIP + PYR that caused the specific oxidative stress disorders. There are plenty of studies that have highlighted the relationship between pesticides and oxidative stress in general [60,61,62]. Furthermore, oxidative stress, as a result of exposure to pesticides, has been observed in various fish species such as rohu (*Labeo rohita),* ranching mahseer (*Tor putitora),* wolf fish (*Hoplias malabaricus),* bluegill (*Lepomis macrochirus),* African catfish (*Clarias gariepinus),* and Nile tilapia (*Oreochromis niloticus)* [2,63,64,65]. Oxidative stress is described as an imbalance between oxidants and antioxidants that usually ends with increased levels of ROS [66]. Most of the time, the damage caused by ROS is a side effect. 

The occurrence of oxidative stress triggered by the presence of PYR was recently mentioned in the study of Azevedo et al. who reported that the main targets of PYR action are NADH dehydrogenase and succinate dehydrogenase, known to be involved in the mitochondrial respiratory chain [31]. These damages could be reflected in the malfunctioning of the mitochondria and, as a result, the significant increase in the production of reactive oxygen species (ROS). Similarly, Maharajan et al. obtained increased levels of ROS, lipid peroxidation, and nitric oxide in zebrafish larvae after 96 h of exposure to 1.66 µg mL^−1^ PYR. Moreover, at the same dose, the impact of PYR was validated for enzymes recognized as part of the antioxidant system, with reduced levels of SOD, CAT, glutathione peroxidase (GPx), and reduced glutathione (GSH) [40]. Ferreira et al. demonstrated that the administration of 15, 25, and 50 mg kg^−1^ FIP in female mice can affect the structure and function of liver cells with changes in the shape and size of hepatocytes and the appearance of steatosis marked by the accumulation of lipid droplets [21]. Similar data were recorded by De Oliveira et al. on mouse liver cells [35]. FIP can also promote cell death; chromatin composition and distribution in hepatocyte nuclei have been shown to be altered after the presence of FIP, leading to chromatin condensation and marginalization as signals for this process [21]. 

It has also been reported that FIP can promote the occurrence of oxidative stress, inflammation, and even neuronal apoptosis [36,67,68,69]. For example, the antioxidant system of the stingless bee (*Partamona helleri*) was triggered by 0.28 ng μL^−1^ FIP leading to increased SOD, CAT, and glutathione S-transferase [68]. Oral administration for 28 days of 2.5, 5, and 10 mg kg^−1^ body weight (bw) FIP to mice resulted in decreased SOD and CAT activity and in increased levels of lipid peroxidation found in the brain and kidney [36]. These aforementioned repercussions were obtained by Wu et al. in a zebrafish animal model after exposure to 0.5, 1, and 2 mg L^−1^ FIP [69]. 

Several studies suggest that acute exposure to FIP has neurotoxic effects on zebrafish embryonic, larval, and adult brains, including disturbances in sensory and motor systems, altered behavior, and changes in gene expression and enzyme activity. A study by Park et al. showed that the accumulation of FIP in aquatic environments affects essential processes during the early developmental stage of zebrafish [70]. Stehr et al. showed in their developmental neurotoxicity study in zebrafish embryos and larvae that FIP affects the development of spinal locomotor pathways in fish by inhibiting a structurally related glycine receptor subtype [71]. 

The present data revealed leukocyte infiltration in different regions of the zebrafish brain after exposure to FIP + PYR mixtures. This process is known to be correlated with the immune response that occurs and facilitates the entry of several effector cells to act in injury, infection, and cellular stress [72]. In addition, leukocyte infiltration begins as a deleterious complication of oxidative stress [73] and has additionally been linked to non-CNS neurological disorders (e.g., anxiety and cognitive impairment) in peripheral inflammatory diseases [74]. There are several investigations that have found the involvement of proinflammatory cytokines in mediating neurological diseases induced by systemic inflammation [75]. Wu et al. observed that exposure to FIP caused oxidative stress, inflammation, and apoptosis in the brain tissue of adult zebrafish, leading to impaired sensory and motor systems and reduced survival rates [69]. 

The ability of FIP to induce oxidative stress was also demonstrated with a fungicide mixture (150 µg L^−1^) tested in the zebrafish model for seven days. This exposure resulted in significant perturbations in locomotor activity by reducing the total distance traveled and the number of crossings in the novel tank test, as well as increased CAT activity, but no changes in lipid peroxidation [76]. 

Therefore, the changes observed after the histopathological analysis of treated adult zebrafish support the changes observed for locomotor activity and the anxiety state. This may explain some of the behavioral findings. Both compounds act predominantly on the CNS but the action is general. Important information was obtained regarding the toxicity of the mixture of FIP and PYR and comparative data can be found in Table 8.

## 5. Conclusions

The results presented indicate that fipronil and pyriproxyfen at different doses can lead to the disruption of locomotor activity and might support anxious behavior along with histological alterations. Therefore, the study highlights the importance of knowing the possible effects determined by this mixture on organisms. Finally, there is more to discover about the interaction of these two compounds, which should be explored in future studies, especially at the molecular level.

## Figures and Tables

**Figure 1 life-13-01874-f001:**
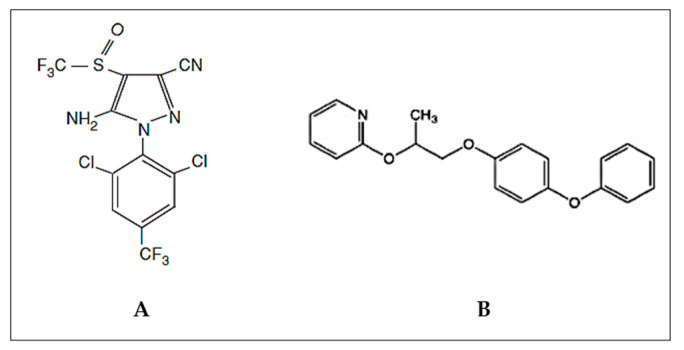
The chemical formula for fipronil (**A**) and pyriproxyfen (**B**).

**Figure 2 life-13-01874-f002:**
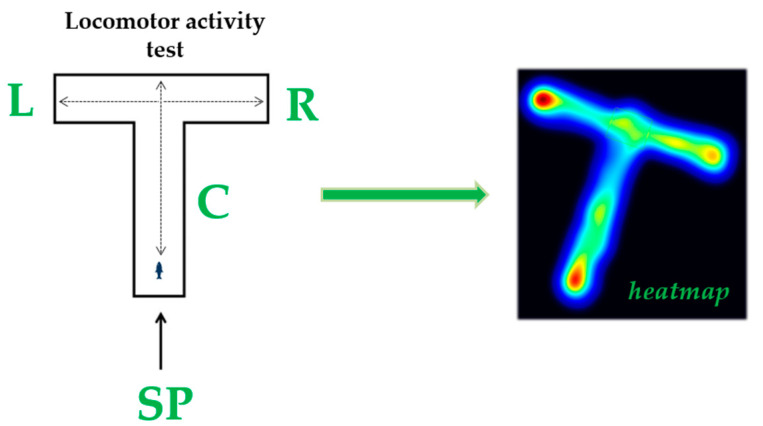
T-maze used to test locomotor activity: organization and heat map (L: left arm, R: right arm, C: center arm, SP: starting point).

**Figure 3 life-13-01874-f003:**
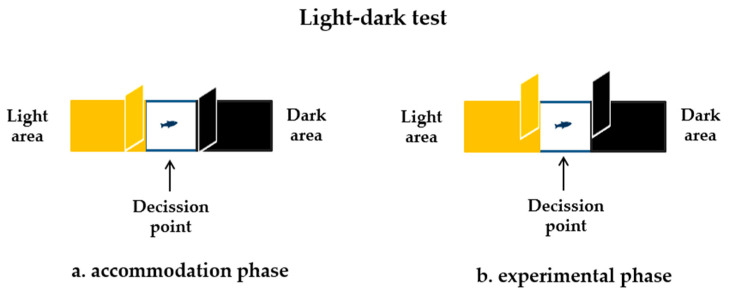
Schematic diagram of the light–dark test adapted to the T-maze and the two steps of behavioral testing (accommodation and experimental phases).

**Figure 4 life-13-01874-f004:**
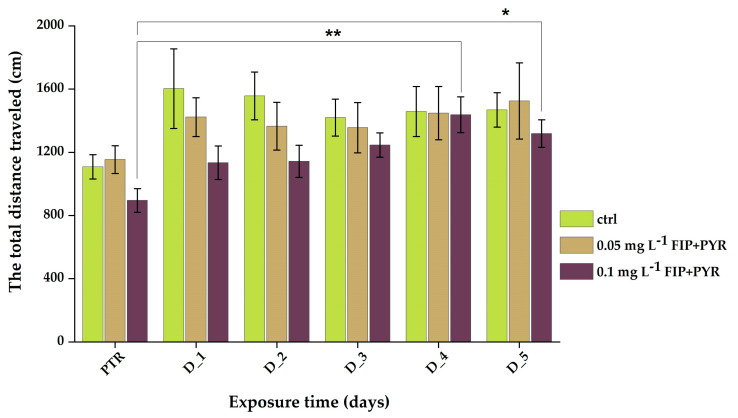
Total distance traveled parameter recorded for all experimental groups during chronic treatment with the pesticide mixture. D stands for day and PTR (pretreatment) is considered as D-0. All values are presented as mean ± S.E.M. (*n* = 10 per group); * *p* < 0.05 and ** *p* < 0.001 ANOVA, Tukey is significant compared to PTR period.

**Figure 5 life-13-01874-f005:**
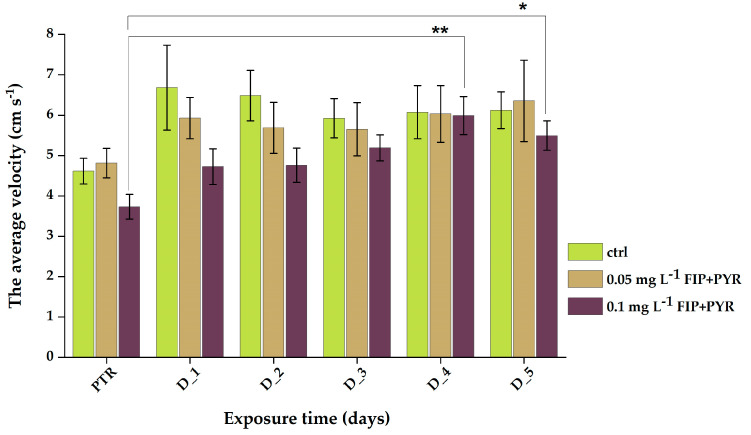
Average velocity parameter recorded for all experimental lots during chronic treatment with the pesticide mixture. D stands for day and PTR (pretreatment) is considered as D-0. All values are presented as mean ± S.E.M. (*n* = 10 per group); * *p* < 0.05 and ** *p* < 0.001 ANOVA, Tukey is significant compared to PTR period.

**Figure 6 life-13-01874-f006:**
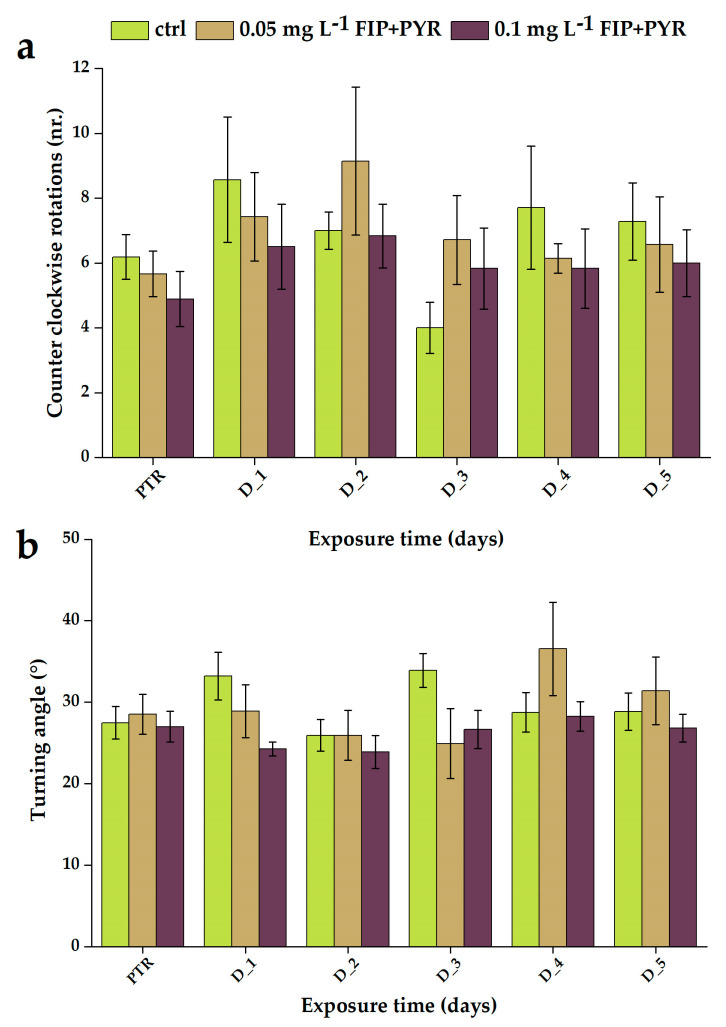
Counterclockwise rotations (**a**) and turning angle (**b**) parameters recorded for all the experimental groups during the chronic treatment with the pesticide mixture. D stands for day and PTR (pretreatment) is considered as D-0. All values are presented as mean ± S.E.M. (*n* = 10 per group).

**Figure 7 life-13-01874-f007:**
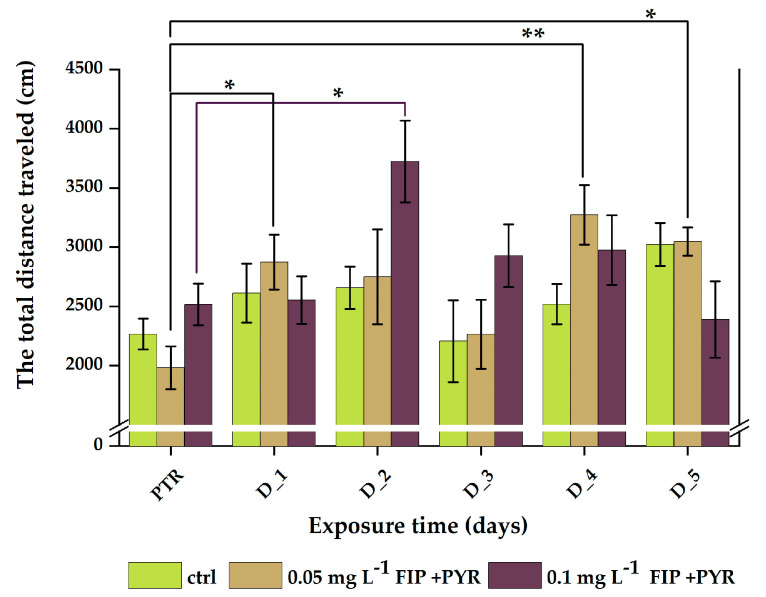
Total distance parameter recorded during the light–dark test for all experimental groups. D stands for day and PTR (pretreatment) is considered as D-0. All values are presented as mean ± S.E.M. (*n* = 10 per group); * *p* < 0.05 and ** *p* < 0.001 ANOVA, Tukey is significant compared to PTR period.

**Figure 8 life-13-01874-f008:**
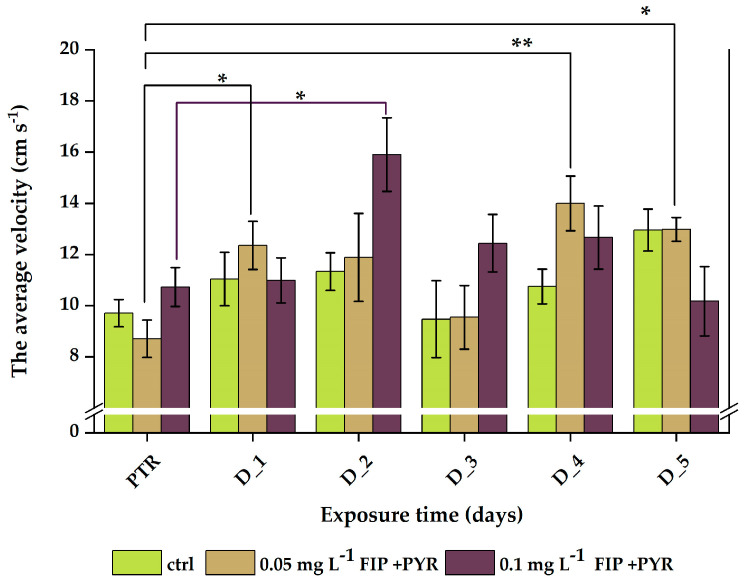
Average velocity parameter recorded during the light–dark test for all experimental batches. D stands for a day and PTR (pretreatment) is considered as D-0. All values are presented as mean ± S.E.M. (*n* = 10 per group); * *p* < 0.05 and ** *p* < 0.001 ANOVA, Tukey is significant compared to the PTR period.

**Figure 9 life-13-01874-f009:**
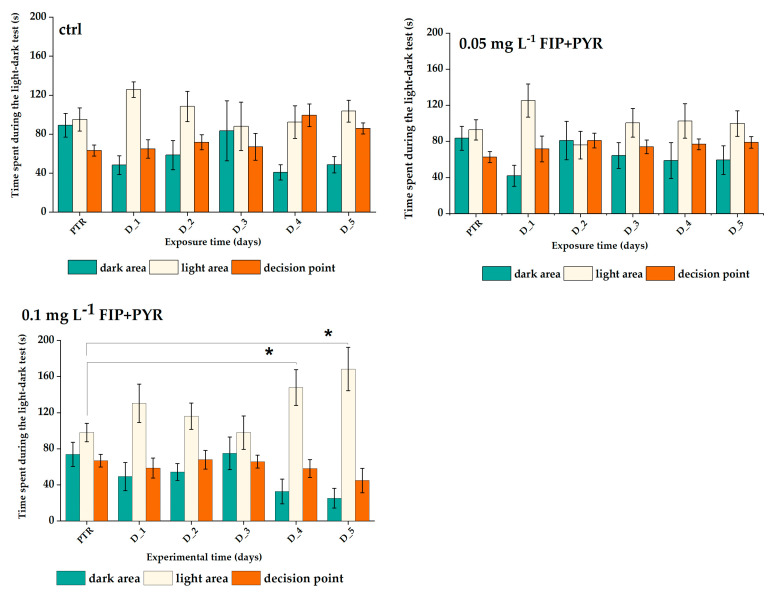
The time spent by the experimental groups during the light–dark test. D stands for a day and PTR (pretreatment) is considered as D-0. All values are presented as mean ± S.E.M. (*n* = 10 per group); * *p <* 0.05 ANOVA, Tukey is significant compared to the PTR period.

**Figure 10 life-13-01874-f010:**
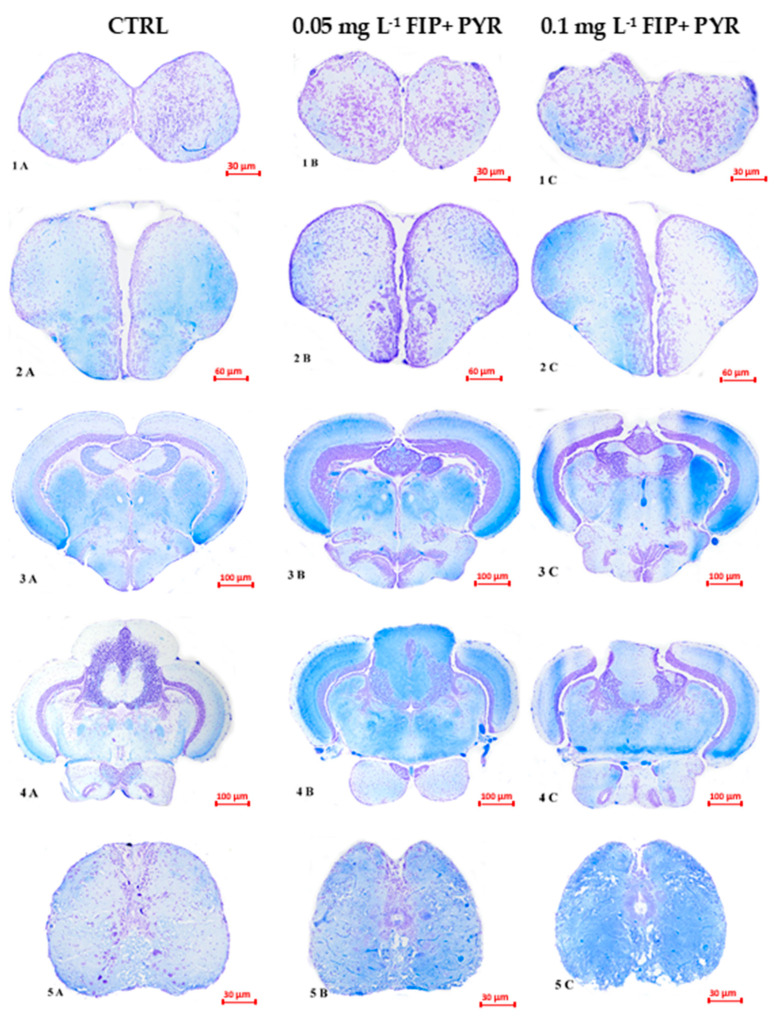
Histological cross-sections from most relevant regions of the control (**left**) and treated (**center** and **right**) zebrafish central nervous system. (**1A**) → (**5A**): Sections from the control group ((**1A**)—olfactory bulbs, scale bar = 30 μm; (**2A**)—telencephalon, scale bar = 60 μm; (**3A**)—diencephalon and mesencephalon, scale bar = 100 μm; (**4A**)—rhombencephalon, scale bar = 100 μm; (**5A**)—medulla spinalis, scale bar = 30 μm). (**1B**) → (**5B**): Sections from the 0.05 mg L^−1^ FIP + PYR group ((**1B**)—olfactory bulbs, scale bar = 30 μm; (**2B**)—telencephalon, scale bar = 60 μm; (**3B**)—diencephalon and mesencephalon, scale bar = 100 μm; (**4B**)—rhombencephalon, scale bar = 100 μm; (**5B**)—medulla spinalis, scale bar = 30 μm). (**1C**) → (**5C**): Sections from the 0.1 mg L^−1^ FIP + PYR group ((**1C**)—olfactory bulbs, scale bar = 30 μm; (**2C**)—telencephalon, scale bar = 60 μm; (**3C**)—diencephalon and mesencephalon, scale bar = 100 μm; (**4C**)—rhombencephalon, scale bar = 100 μm; (**5C**)—medulla spinalis, scale bar = 30 μm). Luxol-Fast Blue—Cresyl Violet staining, ×400.

**Table 1 life-13-01874-t001:** Environmental conditions from the housing and experimental tanks.

Type of Tank	Temperature (°C)	pH	Salinity	Conductivity (µS cm^−1^)	Ammonia (mg L^−1^)
Housing tank	26 ± 0.5	7.6	0.26	551	0.05
Experimental tank	25 ± 0.5	7.4	0.25	553	0.04

**Table 2 life-13-01874-t002:** Data for the total distance traveled parameter recorded for all the experimental groups, expressed as mean ± S.E.M.

Experimental Groups			
Time of Exposure (Days)	Ctrl 0 mg L^−1^ FIP + PYR	0.05 mg L^−1^ FIP + PYR	0.1 mg L^−1^ FIP + PYR
PTR	1107.2 ± 77.5	1153.7 ± 87.8	895.2 ± 74.3
D_1	1602.8 ± 252.2	1422.5 ± 123.01	1134.2 ± 106.03
D_2	1556.1 ± 150.8	1364.6 ± 151.4	1142.8 ± 101.6
D_3	1419.4 ± 116.3	1355.6 ± 158.4	1245.5 ± 77.2
D_4	1457.8 ± 158.4	1447.4 ± 168.2	1437.5 ± 112.9
D_5	1467.9 ± 109.3	1524.1 ± 241.03	1318.3 ± 87.06

**Table 3 life-13-01874-t003:** Data for the average velocity parameter recorded for all the experimental groups, expressed as mean ± S.E.M.

Experimental Groups			
Time of Exposure (Days)	Ctrl 0 mg L^−1^ FIP + PYR	0.05 mg L^−1^ FIP + PYR	0.1 mg L^−1^ FIP + PYR
PTR	4.6 ± 0.3	4.8 ± 0.4	3.7 ± 0.3
D_1	6.7 ± 1.1	5.9 ± 0.5	4.7 ± 0.4
D_2	6.5 ± 0.6	5.7 ± 0.6	4.8 ± 0.4
D_3	5.9 ± 0.5	5.7 ± 0.7	5.2 ± 0.3
D_4	6.1 ± 0.7	6.0 ± 0.7	6.0 ± 0.5
D_5	6.1 ± 0.5	6.4 ± 1.0	5.5 ± 0.4

**Table 4 life-13-01874-t004:** Data for the counterclockwise rotations (A) and turning angle (B) parameters recorded for all the experimental groups, expressed as mean ± S.E.M.

A	Experimental Groups		
Time of Exposure (Days)	Ctrl 0 mg L^−1^ FIP + PYR	0.05 mg L^−1^ FIP + PYR	0.1 mg L^−1^ FIP + PYR
PTR	6.2 ± 0.7	5.7 ± 0.7	4.9 ± 0.9
D_1	8.6 ± 1.9	7.4 ± 1.4	6.5 ± 1.3
D_2	7.0 ± 0.6	9.1 ± 2.3	6.8 ± 1.0
D_3	4.0 ± 0.8	6.7 ± 1.4	5.8 ± 1.2
D_4	7.7 ± 1.9	6.1 ± 0.5	5.8 ± 1.2
D_5	7.3 ± 1.2	6.6 ± 1.5	6.0 ± 1.0
**B**	**Experimental Groups**		
**Time of Exposure (Days)**	**Ctrl** **0 mg L^−1^** **FIP + PYR**	**0.05 mg L^−1^** **FIP + PYR**	**0.1 mg L^−1^** **FIP + PYR**
PTR	2.7 ± 0.2	2.9 ± 0.2	2.7 ± 0.2
D_1	3.3 ± 0.3	2.9 ± 0.3	2.4 ± 0.1
D_2	2.6 ± 0.2	2.6 ± 0.3	2.4 ± 0.2
D_3	3.4 ± 0.2	2.5 ± 0.4	2.7 ± 0.2
D_4	2.9 ± 0.2	3.7 ± 0.6	2.8 ± 0.2
D_5	2.9 ± 0.2	3.1 ± 0.4	2.7 ± 0.2

**Table 5 life-13-01874-t005:** Data for the total distance parameter recorded during the light–dark test for all experimental groups, expressed as mean ± S.E.M.

Experimental Groups			
Time of Exposure (Days)	Ctrl 0 mg L^−1^ FIP + PYR	0.05 mg L^−1^ FIP + PYR	0.1 mg L^−1^ FIP + PYR
PTR	2266.3 ± 130.0	1982.7 ± 180.7	2515.8 ± 175.1
D_1	2612.4 ± 248.3	2874.7 ± 231.1	2553.0 ± 201.4
D_2	2656.0 ± 178.2	2750.2 ± 400.6	3723.5 ± 345.5
D_3	2206.2 ± 345.4	2265.3 ± 291.6	2927.5 ± 264.7
D_4	2519.4 ± 170.5	3273.1 ± 252.3	2974.8 ± 294.7
D_5	3022.2 ± 181.4	3046.8 ± 119.6	2388.9 ± 323.4

**Table 6 life-13-01874-t006:** Data for the average velocity parameter recorded during the light–dark test for all experimental groups, expressed as mean ± S.E.M.

Experimental Groups			
Time of Exposure (Days)	Ctrl 0 mg L^−1^ FIP + PYR	0.05 mg L^−1^ FIP + PYR	0.1 mg L^−1^ FIP + PYR
PTR	9.7 ± 0.5	8.7 ± 0.7	10.7 ± 0.8
D_1	11.0 ± 1.0	12.4 ± 0.9	11.0 ± 0.9
D_2	11.3 ± 0.7	11.9 ± 1.7	15.9 ± 1.4
D_3	9.5 ± 1.5	9.5 ± 1.2	12.4 ± 1.1
D_4	10.7 ± 0.7	14.0 ± 1.1	12.7 ± 1.2
D_5	13.0 ± 0.8	13.0 ± 0.5	10.2 ± 1.4

**Table 7 life-13-01874-t007:** Data for the time spent in the different maze areas recorded during the light–dark test for all experimental groups, expressed as mean ± S.E.M.

		Maze Areas	Time of Exposure (Days)					
			PTR	D_1	D_2	D_3	D_4	D_5
**Experimental Groups**	**Ctrl** **0 mg L^−1^** **FIP + PYR**	**dark**	89.1 ± 12.1	48.3 ± 9.5	58.5 ± 15.1	83.3 ± 30.8	40.7 ± 7.8	48.6 ± 8.3
		**light**	95.0 ± 11.8	125.8 ± 8.0	108.5 ± 15.6	88.0 ± 24.8	92.3 ± 16.7	103.6 ± 11.2
		**decision point**	63.3 ± 5.7	64.9 ± 9.5	71.7 ± 7.8	67.0 ± 13.8	99.4 ± 11.5	85.9 ± 5.7
	**0.05 mg L^−1^** **FIP + PYR**	**dark**	83.5 ± 13.4	41.9 ± 11.7	81.0 ± 21.2	64.3 ± 14.2	58.9 ± 19.7	59.3 ± 16.0
		**light**	92.8 ± 11.2	125.3 ± 18.3	76.0 ± 15.4	100.6 ± 15.8	102.6 ± 19.1	99.8 ± 14.2
		**decision point**	62.6 ± 6.1	71.6 ± 14.3	81.1 ± 8.2	74.0 ± 7.7	76.8 ± 6.0	79.0 ± 6.4
	**0.1 mg L^−1^** **FIP + PYR**	**dark**	73.9 ± 13.4	49.3 ± 15.6	54.2 ± 9.5	75.1 ± 18.1	32.7 ± 13.7	25.2 ± 11.0
		**light**	98.1 ± 10.3	130.5 ± 21.3	116.0 ± 14.5	98.0 ± 18.6	147.9 ± 19.9	168.4 ± 24.1
		**decision point**	66.9 ± 7.0	58.6 ± 11.1	67.9 ± 10.3	65.8 ± 7.2	58.0 ± 9.9	44.9 ± 13.5

**Table 8 life-13-01874-t008:** The comparative effects of FIP and PYR and their mixture on organisms.

Compound	Species	Dose & Time	Effect	Reference
FIP	mussels *Unio delicatus*	0.264 and 0.528 mg L^−1^ 48 h / 7 days	alterations in gills and digestive tissues necrosis	[20]
	zebrafish *Danio rerio*	0.5, 1, and 2 mg L^−1^ 96 h	↓ SOD and CAT activity ↑lipid peroxidation	[69]
		2.5, 7.5, and 15 mg L^−1^ 72 h	↓cell proliferation ↓ hatchability edema and irregular heartbeat	[70]
		0.4 and 0.8 mg L^−1^ 5 days	anxiety swimming performance perturbed ↑ lipid peroxidation	[67]
		0.33 mg L^−1^ 5 days	notochord degeneration locomotor defects	[71]
	common carp *Cyprinus carpio*	0.02 to 0.10 mg L^−1^ 12 days	alterations in gills, liver, kidneys, and intestine tissues	[39]
		0.65 mg L^−1^ 90 days	↓SOD and CAT activity ↑lipid peroxidation	[22]
	mice *Mus musculus*	10 mg kg^−1^ 43 weeks	hyperactivity	[54]
		2.5, 5, and 10 mg kg^−1^ 28 days	↓ SOD and CAT activity ↑lipid peroxidation	[51]
		15, 25, and 50 mg kg^−1^ single administration	steatosis ↑lipid droplets ↑cell death	[21]
		15, 25, and 50 mg kg^−1^	↑steatosis ↑ Kupffer cells alterations of hepatocytes nuclei	[35]
	rat *Rattus norvegicus*	4.85 mg kg^−1^ 6 weeks	impairments in spatial learning and memory ↓dopamine ↓MDA	[18]
		70, 140, and 280 mg/kg	↑ freezing, grooming, and rearing behaviors	[17]
	bobwhite quail *Colinus virginianus*	37.5 mg kg^−1^ 8, 24, 48, 72, and 96 h	↓ food intake and weight gain	[19]
PYR	zebrafish *Danio rerio*	0.33 and 92.5 μmol L^−1^	inhibition of acetylcholinesterase activity ↑ ROS	[31]
		0.125, 0.675, and 1.75 mg L^−1^ 96 h	no locomotor impairments no anxiety-like behavior	[56]
		1.66 µg mL^−1^ 96 h	↑ ROS ↑ lipid peroxidation ↑ nitric oxide ↓ SOD, CAT, GPx, GSH hyperemia pericardial edema	[40]
	platy *Xiphophorus maculatus*	2.5 and 5 µg L^−1^	↓body weight	[30]
	mice *Mus musculus*	30, 100, 300, and 1000 mg kg^−1^	↑ glial cells number, vacuolization, and degenerated neurons	[41]
		100, 200, 320, 600, and 1200 mg kg^−1^ 28 days	↓body weight	[28]
	rat *Rattus norvegicus*	1000, 2000, and 4000 mg kg^−1^	↓body weight	[29]
FIP + PYR	zebrafish *Danio rerio*	**0.05 and 0.1 mg L^−1^** **5 days**	**↑locomotor activity** **↑anxiety level** **↑hemosiderin deposits** **dilated blood vessels and leucocyte infiltration in different brain sections**	**Present work**
		600 µg L^−1^ 14 days	↓swimming performance ↓sociability ↑SOD, GPx, and MDA	[48]

↓—decrease, ↑—increase, SOD—superoxide dismutase, CAT—catalase, MDA—malondialdehyde, ROS—reactive oxygen species, GPx—glutathione peroxidase, GSH—reduced glutathione.

## Data Availability

Data supporting this study cannot be made available due to ethical and commercial reasons.

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
