# Peer review of "Histopathological and Behavioral Impairments in Zebrafish (Danio rerio) Chronically Exposed to a Cocktail of Fipronil and Pyriproxyfen"

_life, 2023, doi:10.3390/life13091874_

Round 1

Reviewer 1 Report

In this manuscript, Robea et al examined the Histopathological and behavioral effects of a cocktail of fipronil (FIP) and pyriproxyfen (PYR). Using zebrafish as a model organism. They claimed FIP and PYR treatments cause behavioral and histological neurological changes. Unfortunately, this manuscript is technical sound report without any mechanistic study of toxicology, which significantly reduce the potential impact of the study. The data presentation is not quite right. The academic writing is not hard to understand and scientific description is in sufficient.

Major points

1.      Why 5 days?.

2.      Comparison should be ctrl, o.o5 and o.1 FIP+PYR. Why compare to D_0?   

3.      Why co-treatment but not single treatment?

4.      Fig.9 and Fig10 should merge to one.

5.      All description of figure legends are insufficient.

Scientific writing is required

Author Response

  1. Why 5 days?

We chose to test this mixture for 5 days based on previously obtained but unpublished data. The treatment regimen can often differ from case to case, considering how this combination is administered to dogs and cats and beyond. According to several published articles, increased efficacy has been reported with a treatment starting from 12 hours to months, depending on the composition and concentration of the mixture (FIP or PYR + other chemicals: Beugnet et al. 2015 - 10.1186/s13071-015-0680-1; Jongejan et al. 2015 - 10.1186/s13071-015-1207-5; Halos et al. 2016 - 10.1186/s13071-016-1345-4; Varloud et al. 2015 - 10.1007/s00436-015-4470-7; Delcombel et al. 2017 - 10.1186/s13071-017-2272-8; Carithers et al. 2018 –

www.jarvm.com/articles/Vol16Iss1/Vol16%20Iss1%20Carithers2.pdf; Rust et al. 2020 - 10.3390/insects11100668; Medication Guides from 2019 - https://cdn.brief.vet/web-files/PVD/drupaluploads/files/Pyriproxyfen%20&%20Pyriproxyfen%20Combination_060619.pdf). Knowing the effect of this mixture is necessary and vital to avoid collateral damage to the health of fauna and flora in addition to environmental imbalance.

  1. Comparison should be ctrl, o.o5 and o.1 FIP+PYR. Why compare to D_0?

In this experiment, D_0 is the pretreatment period. Data were compared for differences between pretreatment and treatment periods for each group. In order to be able to obtain more accurate results, it is recommended to make the comparison between the initial behavior of the organism (considered here as pretreatment) and that observed after exposure to/administration of a chemical compound.

  1. Why co-treatment but no single treatment?

According to the literature, the mixture of fipronil or pyriproxyfen with other compounds is usually recommended in the treatment of dogs and cats (e.g. Beugnet et al. 2015 - 10.1186/s13071-015-0680-1; Jongejan et al. 2015 - 10.1186/s13071-015-1207-5; Halos et al. 2016 - 10.1186/s13071-016-1345-4; Varloud et al. 2015 - 10.1007/s00436-015-4470-7; Delcombel et al. 2017 - 10.1186/s13071-017-2272-8; Carithers et al. 2018 - www.jarvm.com/articles/Vol16Iss1/Vol16%20Iss1%20Carithers2.pdf; Rust et al. 2020 - 10.3390/insects11100668), due to the synergistic effect of the two compounds. Our aim was to investigate the combined effect of FIP and PYR on zebrafish, as they have a similar spectrum of action that increases the impact on organisms and the environment.

  1. 9 and Fig.10 should merge to one.

As suggested by the reviewer, we combined Figures 9 and 10 into one (now Figure 10).

  1. All description of figure legends are insufficient.

Figure legends have been completed as suggested by the reviewer:

”Figure 4. Total distance traveled parameter recorded for all experimental groups during chronic treatment with the pesticide mixture. PTR (pretreatment) is considered as D_0). All values are presented as mean ± S.E.M. (n= 10 per group); *p<0.05 and **p<0.001 ANOVA, Tukey is significant compared to PTR pretreatment period.” in comparison to: ” Figure 3. Total distance traveled parameter recorded for all experimental groups during chronic treatment with the pesticide mixture. All values are presented as mean ± S.E.M. (n= 10 per group); *p<0.05 and **p<0.001 ANOVA, Tukey is significant compared to pretreatment period.”

Same modification for Figures 5-9, previously noted as 4-8.

Reviewer 2 Report

An article by Madalina-Andreea Robea and colleagues reported on the effect of a cocktail of fipronil and pyriproxyfen on histopathological and behavioral impairments of Danio rerio. The authors stated that that the mixture of fipronil and pyriproxyfen have considerable consequences on zebrafish and promote functional neurological and histological changes. While the study is rather interesting, but in my opinion it has several problems related to the data representation.

 1.      Table 1, Housing tank temperature was written without SD (as experimental tank).

2.      Lines 239-240, add the used magnification of the light microscope; add bars on figures 9, 10.

3.      Under the figures 3-7, decipher the abbreviation PTR.

4.      Please explain the use of the concept of “baseline behavior” in relation to the histograms in the figures, how it was determined and where can see its position.

5.      Consider placing tables with measured parameters along with SD (figures 3-7) next to histograms - this will allow readers to compare data better than the presented description.

6.      Figure 8 - it is better to compare different areas (light zone, dark zone, and decision point) at different concentrations of mixtures on histograms.

7.      In conclusion, the authors stated that “fipronil and pyriproxyfen at different doses can lead to impaired motor activity…”, but according to your data, there were no significant differences between the control group and the exposure group in terms of locomotor activity parameters (section 3.1). Please explain.

8.      Since the authors considered in this work only the effect of a mixture of fipronil and pyriproxyfen, it is worth adding to the discussion section a comparative table with the available literature data on the effects of individual substances and the data obtained in your article. This will really help to assess the contribution of individual substances to the observed effects.

 Minor comments

1.      Line 71 - for better understanding, decipher in brackets “fiproles”.

2.      It would be better to put line 75 before line 61.

3.      Please, add a figure with fipronil and pyriproxyfen structures to the introduction section. In my opinion, their presence only in GA is not enough.

4.   In the introduction, the abbreviations PYR and FIP have already been introduced, and further in the text, the full spelling of substances should be removed (section 2.2, …).

Minor editing of English language required

Author Response

  1. Table 1, Housing tank temperature was written without SD (as experimental tank).

We added the SD for the housing tank temperature.

  1. Lines 239-240, add the used magnification of the light microscope; add bars on figures 9, 10.

As the reviewer indicated the magnification of the light microscope and the bars for Fig. 9 (Fig. 9 and 10 has been changed in one figure – Fig. 10) has been added: ” Figure 10. Histological cross-sections from most relevant regions of the control (left) and treated (center and right) zebrafish central nervous system. 1 A 5 A: Sections from the control group (1 A – olfactory bulbs, scale bar = 30 μm; 2 A – telencephalon, scale bar = 60 μm; 3 A – diencephalon and mesencephalon, scale bar = 100 μm; 4 A – rhombenceph-alon, scale bar = 100 μm; 5 A – medulla spinalis, scale bar = 30 μm). 1 B 5 B:  Sections from the 0.05 mg L-1 FIP+ PYR group (1 B – olfactory bulbs, scale bar = 30 μm; 2 B – telen-cephalon, scale bar = 60 μm; 3 B – diencephalon and mesencephalon, scale bar = 100 μm; 4 B – rhombencephalon, scale bar = 100 μm; 5 B – medulla spinalis, scale bar = 30 μm). 1 C 5 C: Sections from the 0.1 mg L-1 FIP+ PYR group (1 C olfactory bulbs, scale bar = 30 μm; 2 C telencephalon, scale bar = 60 μm; 3 C diencephalon and mesencephalon, scale bar = 100 μm; 4 C - rhombencephalon, scale bar = 100 μm; 5 C medulla spinalis, scale bar = 30 μm). Luxol-Fast Blue – Cresyl Violet staining, x400.

  1. Under the figures 3-7, decipher the abbreviation PTR.

We made the corrections suggested for the figures 4-9 (previously 3-8); for instance: ”Figure 4 (previously Fig. 3). Total distance traveled parameter recorded for all experimental groups during chronic treatment with the pesticide mixture. Pretreatment (PTR) is considered as D_0). All values are presented as mean ± S.E.M. (n= 10 per group); *p<0.05 and **p<0.001 ANOVA, Tukey is significant compared to PTR pretreatment period.

  1. Please explain the use of the concept of “baseline behavior” in relation to the histograms in the figures, how it was determined and where can see its position.

Just to observe the unaltered behavior we performed a series of tests before exposure to the chemical compounds; the results were integrated in the form of the initial behavior. All the results of the controlled intoxication were compared with the initial ones to observe the existence of differences clearly and concisely between the two phases of the study: pretreatment vs. treatment. Thus, the pretreatment phase was considered as initial behavior which is abbreviated in histograms as PTR.

  1. Consider placing tables with measured parameters along with SD (figures 3-7) next to histograms - this will allow readers to compare data better than the presented description.

Following the reviewer’s suggestion, we completed the graphical representations of measured parameters with corresponding tables; in the manuscript starting with the table 2 until table 7  (eg.” Table 2. Data for the total distance traveled parameter recorded for all the experimental groups expressed as mean ± S.E.M.”).

  1. Figure 8 - it is better to compare different areas (light zone, dark zone, and decision point) at different concentrations of mixtures on histograms.

The data from the present study were compared to observe a possible difference between the initial behavior and the behavior recorded after exposure to the chemical compounds for each group. We believe that this option of data analysis and interpretation is more suitable for extracting more information than comparing the tested groups with each other. Of course, this can be done, but the variability of the answers will be much greater.

  1. In conclusion, the authors stated that “fipronil and pyriproxyfen at different doses can lead to impaired motor activity…”, but according to your data, there were no significant differences between the control group and the exposure group in terms of locomotor activity parameters (section 3.1). Please explain.

Tukey's test showed significant differences for all parameters investigated between the two phases of the experiment: the pretreatment considered as baseline behavior and the treatment phase – the actual exposure to the toxic mixture, for each group (see Figures 4-9). Only for counterclockwise rotations and turning angle (Fig. 6), Tukey's test did not show significant differences.

  1. Since the authors considered in this work only the effect of a mixture of fipronil and pyriproxyfen, it is worth adding to the discussion section a comparative table with the available literature data on the effects of individual substances and the data obtained in your article. This will really help to assess the contribution of individual substances to the observed effects.

A comparative table with the available literature data on the effects of individual substances (fipronil and pyriproxyfen) and data obtained in the present manuscript was added in this manuscript (Table 8. The comparative effects of FIP, PYR and its mixture on organisms).

  1. Line 71 - for better understanding, decipher in brackets “fiproles”.

As suggested, we added in brackets the meaning of fiproles: ”... for 12 US facilities, the discharge from the treatment plant reported fiproles (FIP and its degradation products) concentrations ranging…” (line 74).

  1. It would be better to put line 75 before line 61.

The line 75 was moved before the line 61 as suggested.

  1. Please, add a figure with fipronil and pyriproxyfen structures to the introduction section. In my opinion, their presence only in GA is not enough.

The figure with the two chemical structures was added as Figure 1.

  1. In the introduction, the abbreviations PYR and FIP have already been introduced, and further in the text, the full spelling of substances should be removed (section 2.2, …).

We corrected allover within the text.

Round 2

Reviewer 1 Report

The authors' responses is sufficient.

Author Response

Thank you for the fruitful comments!

Reviewer 2 Report

The revised version of the article presents tha data well and looks more represantative than the original draft. The authors responded to all my comments.

Minor: linews 336, 358, 536 - "table" should be capitalized 

Author Response

Thank you for the fruitful comments!